# Future Frame Synthesis for Fast Monte Carlo Rendering

Zhan Li*
Portland State University

Carl S Marshall†
Intel

Deepak S Vembar ‡
Intel

Feng Liu§
Portland State University

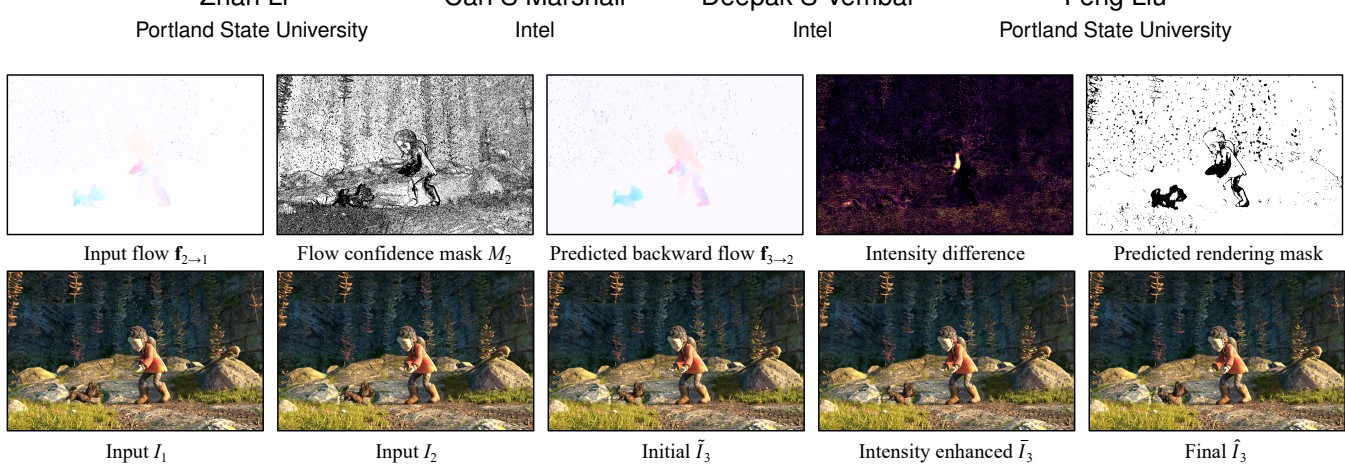

Input flow $\mathbf{f}_{2\rightarrow 1}$    Flow confidence mask $M_2$    Predicted backward flow $\mathbf{f}_{3\rightarrow 2}$    Intensity difference    Predicted rendering mask

Input $I_1$    Input $I_2$    Initial $\tilde{I}_3$    Intensity enhanced $\bar{I}_3$    Final $\hat{I}_3$

Figure 1: Given two input frames $I_1$ and $I_2$ together with the optical flows between them $\mathbf{f}_{2\rightarrow 1}$ and the flow confidence map $M_2$, our method first estimates backward flows $\mathbf{f}_{3\rightarrow 2}$ and uses it to generate an initial future frame $\tilde{I}_3$. Our method then predicts the intensity difference map to compensate for the pixel-wise intensity difference as the intensity values of corresponding pixels could potentially change over time. Finally, our method predicts a backward flow confidence map, uses it to calculate a rendering map to optionally select those unreliably predicted pixels to re-render using an off-the-shelf rendering engine.

## ABSTRACT

Monte Carlo rendering algorithms can generate high-quality images; however they need to sample many rays per pixel and thus are computationally expensive. In this paper, we present a method to speed up Monte Carlo rendering by significantly reducing the number of pixels that we need to sample rays for. Specifically, we develop a neural future frame synthesis method that quickly predicts future frames from frames that have already been rendered. In each future frame, there are pixels that cannot be predicted correctly from previous frames in challenging scenarios, such as quick camera motion, object motion, and large occlusion. Therefore, our method estimates a mask together with each future frame that indicates the subset of pixels that need ray samples to correct the prediction results. To train and evaluate our neural future frame synthesis method, we develop a large ray-tracing animation dataset. Our experiments show that our method can significantly reduce the number of pixels that we need to render while maintaining high rendering quality.

**Index Terms:** Computing methodologies—Computer graphics—Ray tracing

## 1 INTRODUCTION

Monte Carlo ray tracing algorithms are widely used to generate photorealistic images for many applications, such as computer games, films, and simulations. However, these algorithms are time-consuming as they need to sample many rays to shade each pixel [9, 54].

A great amount of effort has been devoted to fast Monte Carlo rendering. A popular category of approaches is to only cast a small number of rays for each pixel and then reconstruct a high-quality

---

*e-mail: lizhan@pdx.edu
†e-mail: carl.s.marshall@intel.com
‡e-mail: deepak.s.vembar@intel.com
§e-mail: fliu@cs.pdx.edu

rendering from these few samples by denoising [8, 13, 20, 21]. Another category of approaches is to first reproject rays sampled when rendering previous frames to the current frame and use them to reconstruct the current frame [3, 4]. These temporal reprojection methods have difficulty in rendering view-dependent effects and filling pixels that are occluded in the previous frames.

This paper presents a future frame synthesis method for fast Monte Carlo rendering. Our method belongs to the category of reprojection algorithms and improves existing algorithms by exploiting deep neural networks to synthesize a future frame from frames that have already been rendered. Existing reprojection methods use forward warping to splat samples / pixel colors from previous frames to the future frame, which often suffers artifacts such as holes. To achieve higher rendering quality, our method uses backward warping to synthesize the future frame from previous frames. Backward warping, however, requires optical flow from the future frame to the previous frame(s), which cannot be calculated without the future frame or some of its intermediate G-buffer data. To address this problem, we train a deep neural network to learn to predict the backward flow of future frames. As the color constancy assumption for reprojection algorithms may not always hold across neighboring frames, we employ a second neural network to predict the intensity differences from previous frames, which are then added to the synthesized future frame. Furthermore, our future frame synthesis networks may generate errors when facing challenging scenarios, such as large occlusions and significant view-dependent effects. Therefore, as an optional step, our method uses a mask neural network to generate a confidence map that indicates those unreliable pixel estimates and re-render these pixels using ray tracing.

As there are no publicly available large-scale ray tracing animation dataset, we built such a dataset by collecting or purchasing model and scene files and render them using the Unreal Engine or Blender Cycles. Our dataset contains many animation sequences with a variety of animation characters, background scenes and camera motion. This dataset allows us to train and test our neural future frame synthesis method. Our experiments show that our method is able to drastically reduce the number of rays that need to be sampled to produce frames while maintaining high rendering quality.

## 2  RELATED WORK

Fast Monte Carlo rendering has a rich literature history. A popular approach is to reduce the total number of sampling rays that need to be cast to generate an image [54]. A large number of algorithms have been developed that only sample a small number rays per pixel and then perform denoising to reconstruct high-quality renderings [6, 12, 17, 22–24, 29, 34, 36, 46]. The recent learning-based denoising methods, especially those use deep neural networks, can generate very high-quality renderings with only a small number of samples [8, 13, 20, 21, 30].

Another approach is to reuse samples from previous frames by reprojecting the samples to future frames [3, 4, 47, 48]. These reprojected samples are often used together with new samples to reconstruct future frames [10, 39, 41]. Recently, temporal reprojection has been an important step for a denoising method to make use of samples or features from neighboring frames to obtain high-quality denoising results [15, 16, 45]. Besides, reusing temporal rendering information has been widely explored for a variety of other rendering problems. For instance, Scherzer *et al.* reuse past information to reduce the computation cost of shadow mapping [37]. Nehab *et al.* developed a reverse reprojection-based caching scheme that enables pixel shaders to reuse calculations performed for visible surface points over time [32]. Asynchronous time warp reprojects past frames to the future frame to reduce the latency in VR applications [43]. Didyk *et al.* warp existing frames to increase frame rates for high-refresh-rate displays [11]. Yang *et al.* further increased frame rates via bidirectional scene reprojection [50]. Recently, Mueller *et al.* reported that it is possible to apply temporal shading reuse to extended periods of time for a significant portion of samples and demonstrated that for real-time VR applications [31]. Like these methods, our work also explores temporal rendering history to speed up rendering and focuses on Monte Carlo rendering algorithms. Our method learns to predict backward flows that allow for future frame synthesis without the need of hole filling. Moreover, inspired by deep adaptive sampling methods, such as Kuznetsov *et al.* [21], our method predicts a confidence map that can be used to identify unreliable pixels in the predicted future frame and optionally re-render them using an off-the-shelf ray tracing rendering engine.

Our work is also related to deep video frame prediction methods from the Computer Vision community [7, 25, 26, 28, 35, 40, 44, 49, 53]. These methods employ a variety of deep neural network algorithms to learn to predict future frames from their previous video frames. Particularly, given its good performance in predicting future frames, our work adopts the neural network architecture of SDCNet from Reda *et al.* [35] to estimate the backward flows. Unlike Reda *et al.* that use a neural network to estimate the optical flow between the previous frames, our method uses the optical flows from the rendering engine and predicted by our future frame synthesis network. Since optical flows, even from the rendering engine, are not perfect, we further compute or predict a confidence map and feed it to the backward flow estimation network to improve the quality of the backward flows. Moreover, we further improve future frame prediction quality by estimating and compensating for the intensity difference over time and predicting a confidence map to guide the rendering of unreliably predicted pixels.

Finally, in a concurrent work, Guo *et al.* developed ExtraNet, which also extrapolates future frames to achieve low-latency rendering [14]. In addition to fully rendered previous frames, their method renders G-buffer data of the extrapolated frames as input, which allows their method to employ a lightweight network to render high-quality extrapolated frames. In contrast, our method does not need G-buffer data of the extrapolated frames and thus requires less memory consumption. However, without the G-buffer data of the extrapolated frames, our method sometimes cannot predict future frames as high quality as ExtraNet. Nevertheless, future frame prediction is necessarily error-prone even with the target G-buffer data.

Therefore, our method also predicts an error mask that identifies difficult-to-predict pixels and allows a rendering engine to optionally render these pixels to ensure the quality of the final future frames.

## 3  RAY-TRACING ANIMATION DATASETS

### 3.1  Rendering Engines

We use Unreal Engine 4 (UE4) to render our animation dataset. Since the path tracer in UE4 is not stable for production [42], we use its hybrid ray tracer called "Real-Time Ray Tracing" (RTRT). We train and test our future frame synthesis network using the animation sequence rendered by RTRT. To examine how well our network can be generalized to examples generated by a pure path tracer, we also use Blender Cycles to render additional animation sequences and use them to test our network.

### 3.2  Digital Assets

We purchased Unreal scene files from *UE Marketplace* and used each of them as the background for an animation sequence. We obtained animations with characters from *Mixamo* and integrated them into the background scenes to generate various animation sequences. Specifically, we bought 20 background environments from *UE Marketplace*. We separated them into three groups: 10 for training, 1 for validation, and the remaining 9 for testing. For each background environment, we randomly added animation characters. Then we picked good viewpoints and created camera paths to follow the main animation character for each animation scene following a recent method used to create the Creative Flow+ Dataset [38]. In this way, we could generate multiple animation sequences with different camera paths from the same animation scene. We took care to prevent the camera going through animation characters. In total, we produced 119 videos for the training set, where each video has 461 frames. For the validation and testing set, we followed the same approach but only generate one video for each animation scene. Our testing and validation set contains 9 and 1 animation sequences, respectively. When rendering these animation sequences, we individually adjusted the number of samples per pixel to avoid noticeable noise in the results. Samples of our animation sequences are shown in Figure 2.

We also created a second testing set. Specifically, we used Blender Cycles to render 6 animation sequences from resources from the Blender Open Movies dataset [1] and the Nvidia ORCA dataset [27]. When rendering animation sequences from the Blender Open Movies and Nvidia ORCA datasets, we used 1000 samples per pixel and 2000 samples per pixel, respectively.

### 3.3  Ground Truth Optical Flow

We followed Fan *et al.* [18, 19] and the suggestions from the Unreal Community [2] to compute the ground-truth optical flows between two consecutive animation frames. Specifically, we used the Unreal built-in optical flow tool to compute the optical flow of still background scene induced by camera movements. We used texture coordinates to compute optical flows of moving objects. However, we were not able to compute the ground-truth optical flows for several scenarios, such as shadow regions and transparent or semi-transparent objects. Since shadows and semi-transparency are common in ray tracing renderings, we kept them in our dataset with no ground-truth optical flows for them.

## 4  FUTURE FRAME SYNTHESIS

Given two consecutive frames $I_{1:2}$, our method aims to predict a sequence of future frames $\hat{I}_{3:t}$ frame by frame. For instance, we first predict $\hat{I}_3$ from $I_1$ and $I_2$ and then predict $\hat{I}_4$ from $I_2$ and $\hat{I}_3$. Below we describe how our method predicts $\hat{I}_3$. The other future frames are generated in the same way with minor changes that will be noted in this paper.

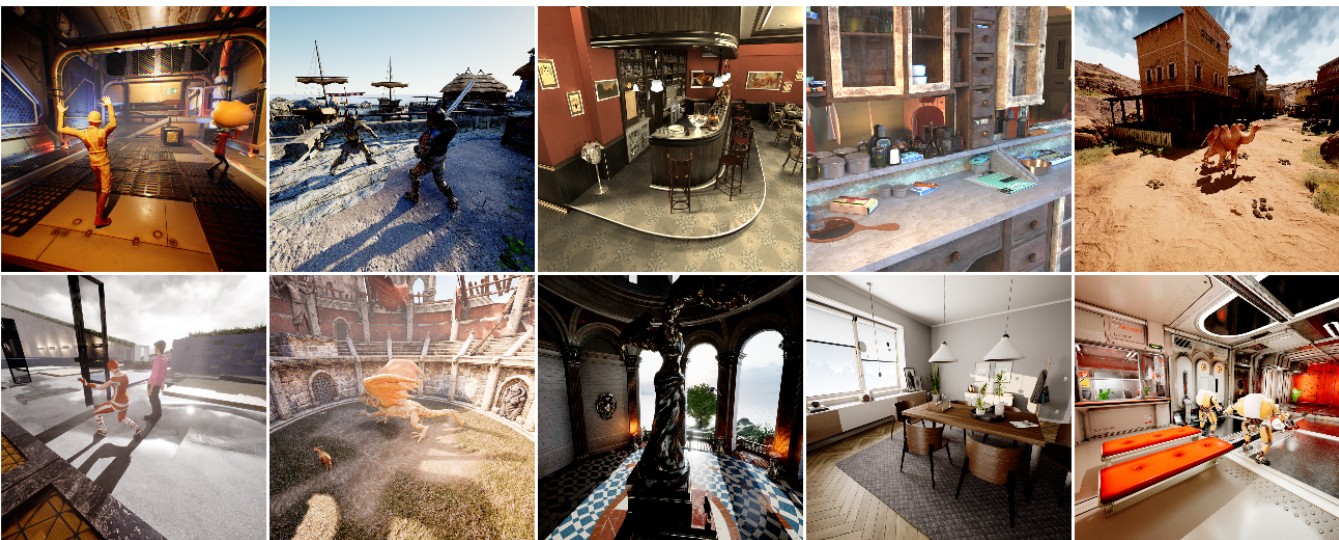

Figure 2: Samples of our ray tracing animation dataset.

We use a deep neural network to predict $\hat{I}_3$. As shown in Figure 3, our network takes as input two existing frames $I_1$, and $I_2$, the optical flow map $f_{2\rightarrow1}$ from $I_2$ to $I_1$, and the optical flow confidence map $M_2$. Following the previous video frame interpolation and extrapolation papers [33, 35], our network outputs the backward flow from $\hat{I}_3$ to $I_2$, denoted as $f_{3\rightarrow2}$, and then uses it to synthesize the future frame $\hat{I}_3$ from $I_2$ by backward warping. Such an approach tends to generate sharper frames than estimating the future frame directly. In this paper, we adopt the network architecture from SDC-Net [35] for backward optical flow estimation.

There are necessarily errors in the predicted future frame $\hat{I}_3$. For example, when the camera angle or camera location changes, content that is invisible in the previous frames will be visible in the future frame. Warping the previous frames cannot generate those disoccluded content in the future frame. Significant view-dependent effects also poses challenges for future frame prediction. Therefore, we added another neural network that shares the same input as the backward flow estimation network to estimate a confidence map $M_3$, as illustrated in Figure 3. This confidence estimation network also shares the same network architecture as the backward flow estimation network with an additional sigmoid layer at the end. Each element in this map indicates how reliable the corresponding optical flow in $f_{3\rightarrow2}$ can be used to estimate the pixel color for the future frame. This confidence map provides an optional step to improve the future frame quality by re-rendering those pixels using the rendering engine in the system. In our experiments, we re-render those pixels with the confidence values below a threshold value $\lambda$.

Note, when estimating $\hat{I}_3$, $f_{2\rightarrow1}$ is directly computed from the rendering engine. As discussed in Section 3, the optical flows from the rendering engine are not perfect in many scenarios. More importantly, even when optical flows that correctly accounts for scene point motions do not lead to perfect a future frame. Occlusion and significant view-dependent effect are two common reasons. Therefore, our method computes a confidence map for optical flows. Particularly, the optical flow confidence map $M_2$ is computed by first backwardly warping $I_1$ to align with $I_2$ using $f_{2\rightarrow1}$ and then thresholding the error map against a constant $\omega$. If the error is smaller than $\omega$, we set the corresponding value in $M_2$ 1, otherwise 0. The default value for $\omega$ is 0.04 in our paper when the pixel value is normalized to the range of [0, 1]. When estimating the other future frames $\hat{I}_t$ with $t > 3$, we use $f_{t\rightarrow t-1}$ and $M_{t-1}$, which are both the output from the previous step to estimate $\hat{I}_{t-1}$.

### 4.1 Intensity Enhancement

Our method described above synthesizes a future frame from its immediate previous frame and thus implicitly assumes the intensity constancy. Such an assumption, however, does not always hold. To address this problem, we employ an intensity enhancement network that estimates the intensity differences from the previous frames. Specifically, our method first warps $I_1$ to align with $I_2$ via backward warping and then calculates the intensity difference map between them $B_{1,2}$. We warp the intensity difference $B_{1,2}$ with estimated optical flow to get initial $B_{2,3}$. Our method then feeds initial $B_{2,3}$ together with the initial future frame $\tilde{I}_3$, which is created by warping $I_2$ using the estimated optical flow $f_{3\rightarrow2}$, into the intensity enhancement network to estimate the intensity difference map $\hat{B}_{2,3}$. Our method finally adds $\hat{B}_{2,3}$ to the initial future frame $\tilde{I}_3$ to generate the enhanced future frame $\bar{I}_3$, as shown in Figure 3.

**Loss functions.** We train our frame synthesis network in an end-to-end fashion by computing the losses from predicting three consecutive frames $\hat{I}_{3:5}$ from two input frames $I_{1,2}$ as follows.

$$\mathscr{L} = \sum_{t=3}^{t=5} \delta_t \mathscr{L}_t, \tag{1}$$

where $\mathscr{L}_t$ is the loss from predicting $\hat{I}_t$, $\delta_3 = 1$, $\delta_4 = 0.5$, $\delta_5 = 0.25$. $\mathscr{L}_t$ has the following three components.

$$\mathscr{L}_t = \alpha \mathscr{L}_{t,l_1} + \beta \mathscr{L}_{t,m} + \gamma \mathscr{L}_{t,re} \tag{2}$$

where $\mathscr{L}_{t,l_1}$ is the $\ell_1$ loss between the ground truth $I_t$ and the synthesized future frame $\hat{I}_t$, $\mathscr{L}_{t,m}$ is the binary cross entropy loss between the predicted confidence mask $M_t$ and the ground-truth confidence mask of the enhanced $\bar{I}_t$, which is obtained by thresholding the error ($\omega = 0.04$) between enhanced $\bar{I}_t$ and ground truth $I_t$, and $\mathscr{L}_{t,re}$ is the percentage of pixels that needed to be rendered. We empirically set $\alpha = 0.3$, $\beta = 0.3$, and $\gamma = 0.3$.

**Implementation details.** We randomly crop training images into patches of size $256 \times 256$ from our training images. We use PyTorch to implement our future frame synthesis network. We use a mini-batch of 4. We use the Adam optimizer with multiple-step learning rates. The learning rate is $10^{-4}$ for the first 250 epochs. Then learning rate is set to $10^{-5}$. We train our networks for 700 epochs using one Nvidia Titan Xp. We use 2D convolution layers with a kernel size of 7 to extract features with channels of 32 from stacked

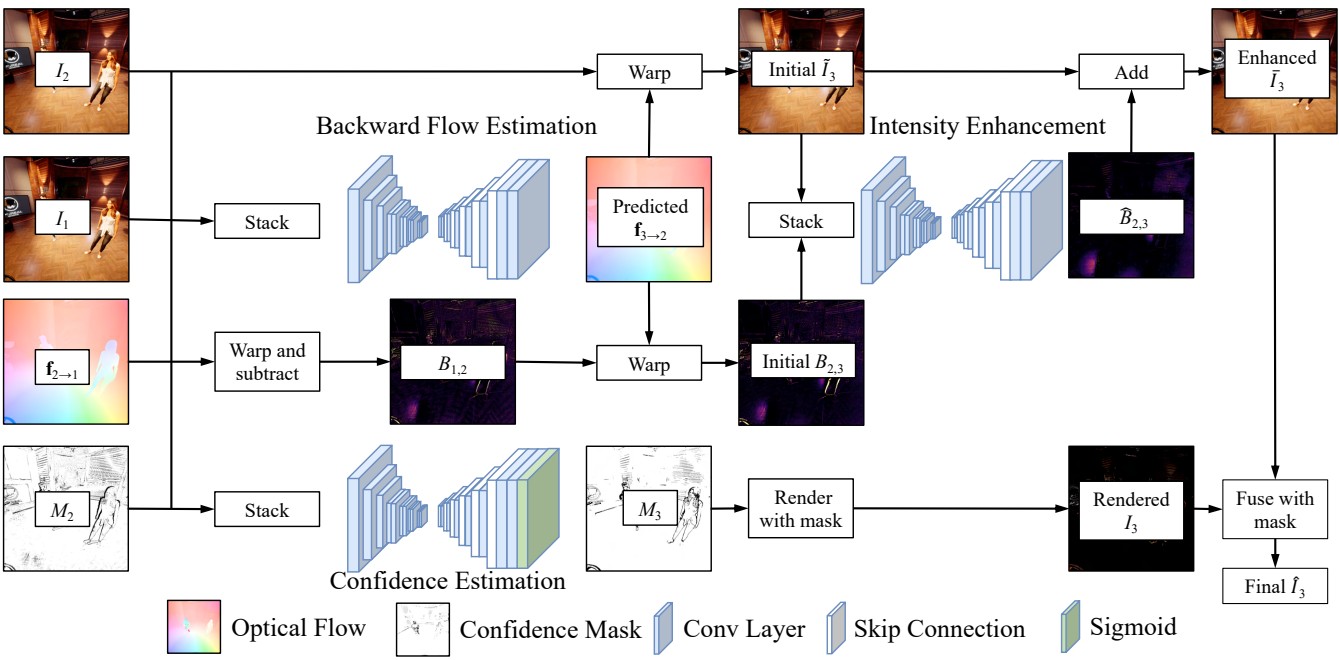

Optical Flow    Confidence Mask    Conv Layer    Skip Connection    Sigmoid

Figure 3: Our future frame synthesis framework.

inputs. The encoders are composed of 5 2D convolution layers using a stride of 2 and 4 2D convolution layers using a stride of 1. The channel numbers increase from 32 to 512 during the encoder. For the decoder, we use 6 2D deconvolution layers with a stride of 2. We use a 2D convolution layer with a stride of 1 and kernel size of 3 to predict the optical flows, masks, and intensity enhancement after the decoders in three networks.

## 5 EXPERIMENTS

We evaluate our method by comparing to baseline reprojection methods and state-of-the-art video frame prediction methods. We also conduct ablation studies to evaluate individual components of our method. In our experiments, we train our future frame synthesis network using the training set of our Unreal animation dataset. We test our method on the corresponding testing set (UE4). As our Unreal dataset was rendered using its hybrid rendering engine, we further test our trained network on our Cycles dataset (Cycles) that was rendered using a ray tracing engine as discussed in Section 3.

### 5.1 Comparisons

**Reprojection methods.** We first compare our method to a baseline reprojection method that warps the current frame $I_2$ to the future frame $I_t$ using forward warping. For such a baseline approach, we first obtain the optical flows from $I_2$ to $I_t$, denoted as $\mathbf{f}_{2 \to t}$. Assuming the linear pixel motion, $\mathbf{f}_{2 \to t}$ can be computed as follows.

$$\mathbf{f}_{2 \to t} = (2-t) * \mathbf{f}_{2 \to 1} \tag{3}$$

where $\mathbf{f}_{2 \to 1}$ is the ground-truth optical flow computed by the rendering engine. We then forward warp $I_2$ to a future frame $I_t$. Multiple pixels could be forwarded to the same target pixel. We blend these pixels in two ways. One is to choose the pixel that is closest to the camera and the other is to blend these pixels using weights that are computed as the inverse of their depth values [5]. We denote them as *reproj-nn* and *reproj-blend* respectively in this section. In addition, forward warping leads to holes in the future frame. We fill these holes using ground-truth pixels from the rendering engine.

As described in Section 4, we use a threshold $\lambda$ to select a subset of pixels to re-render using the rendering engine. Specifically, if the value in the predicted confidence map is smaller than $\lambda$, we

re-render that pixel. Therefore, as we increase the $\lambda$ value, more pixels are re-rendered, as shown in Figure 4. As also shown in the third column of this figure, the mask prediction accuracy of our method also improves as we increase the $\lambda$ value from 0.1 to 0.4. This is because with a small $\lambda$ value like 0.1, our method only selects to a small number of unreliably predicted pixels to re-render while leaving many more unfixed. As we increase the $\lambda$ value, more of those bad pixels are selected to fix. This is also related to the fact that when training our network, we use $\lambda = 0.4$ to compute the mask loss in Equation 2. With $\lambda = 0.4$, our method needs to re-render around 12.5% for Unreal testing examples (UE4) and 10% for Cycles testing examples. While with $\lambda = 0.2$, our method only needs to re-render less than 4.0% for Unreal testing examples (UE4) and 3.0% for Cycles testing examples. Many of the visual examples in this paper are rendered with $\lambda = 0.2$.

As we would expect, the quality of our future frame synthesis method increases as the percentages of re-rendered pixels rises. With a similar amount of re-rendered pixels ($\lambda = 0.2$), our method significantly outperforms the above baseline reprojection approaches in terms of both PSNR ($> 1.5$dB) and LPIPS ($< 0.06$) [52]. These results are consistent on both the Unreal dataset and the Cycles dataset. As shown in Figure 5, *reproj-nn* tends to generate results with aliasing artifacts while the results from *reproj-blend* suffer from the ghosting artifacts. In contrast, our results can predict higher quality future frames.

**Video frame prediction.** To conduct a fair comparison to state-of-the-art video frame prediction approaches, we use our future frame synthesis results without replacing pixels according to the predicted confidence map. For MCNet [44], VoxelFlow [53] and Improvednn [7], we use their official codes. For SDC2D [35], we used the code from released by [51], which is slightly different from their original version SDC-Net in that SDC2D only estimates the backward flows for future frame prediction without estimating the spatially-varying kernels and uses 2D convolutions instead of 3D convolutions. To examine the effect of starting future frame synthesis using the optical flows generated by the rendering engine, we also extended the original SDC2D method by using the rendered optical flows instead of the optical flows estimated from the input frames. We denote this version of SDC2D-GTflow. Note, the rendered

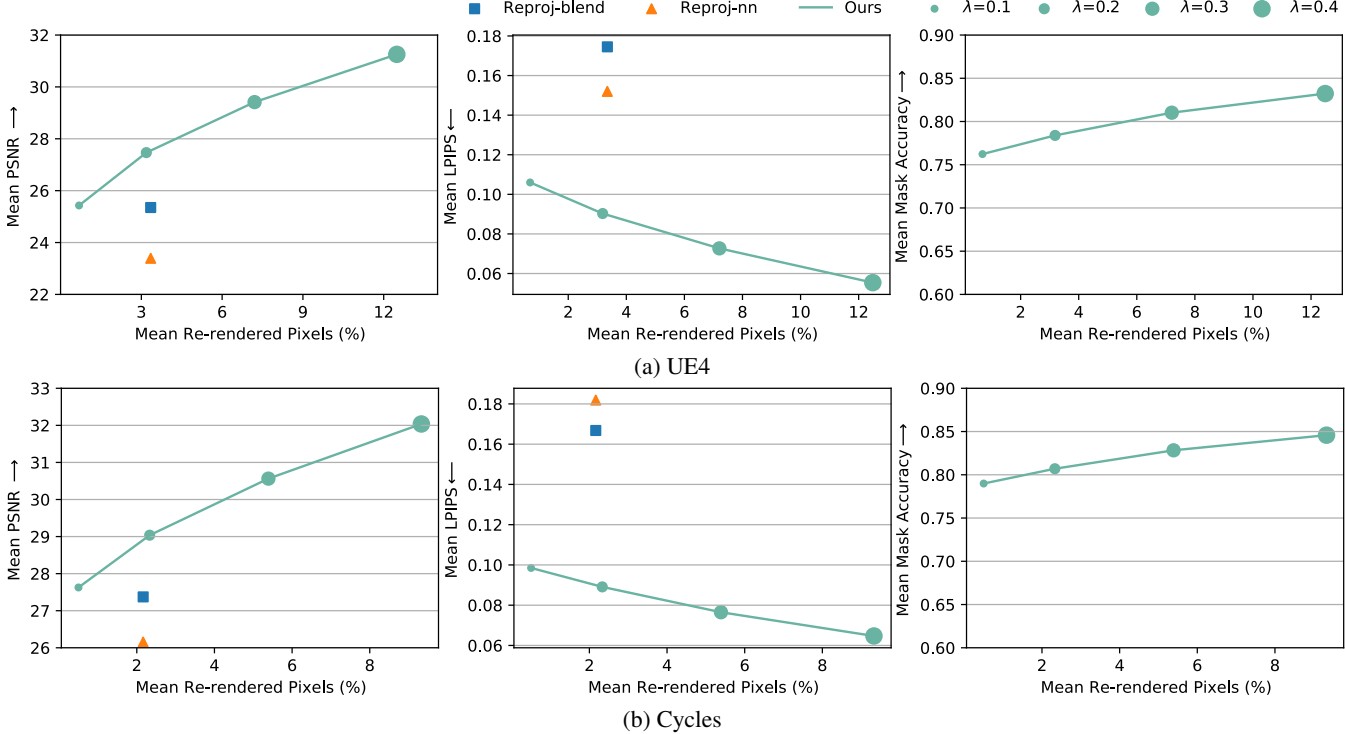

Figure 4: Test results on both UE4 testset (a) and Cycles testset (b). $\lambda$ is the threshold used to select unreliably predicted pixels according to the predicted confidence mask. A larger $\lambda$ selects more pixels to re-render. These results show that our method produces higher-quality renderings in terms of both PSNR and LPIPS while re-rendering a similar amount of pixels to the two baseline reprojection methods.

(a) An example from UE4

(b) An example from Cycles

Overlapped input t=1,2    GT    Reproj-nn    Reproj-blend    Ours-$\lambda$=0.2

Figure 5: Examples of predicting three continuous future frames. We select $\lambda = 0.2$ for our method, which requires similar percentages of re-rendered pixels to other methods.

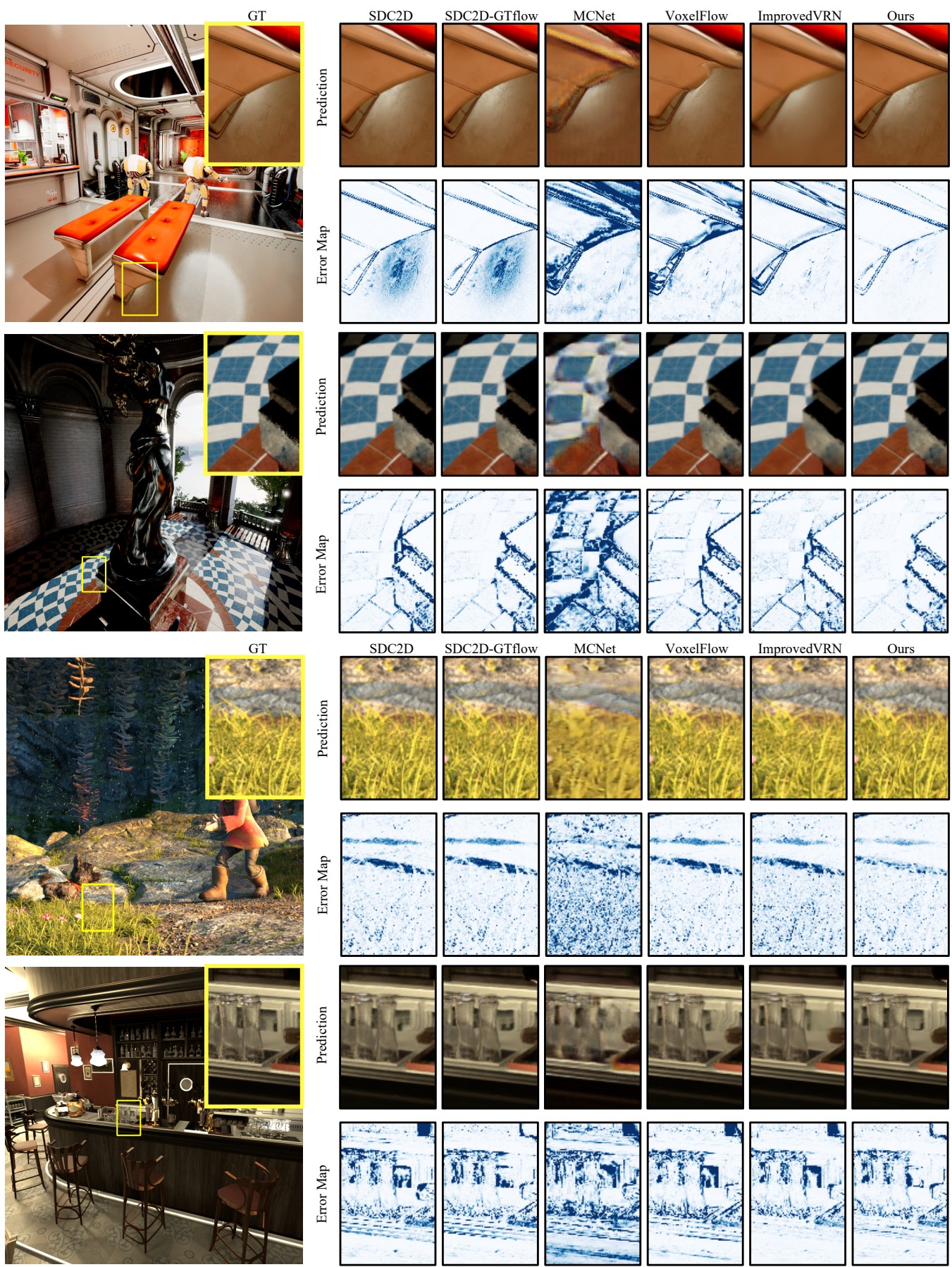

Figure 6: Visual comparisons with video frame prediction methods. The top two examples are from UE4. The bottom two examples are from Cycles. To ensure a fair comparison, we did not re-render pixels in this test when producing our prediction results.

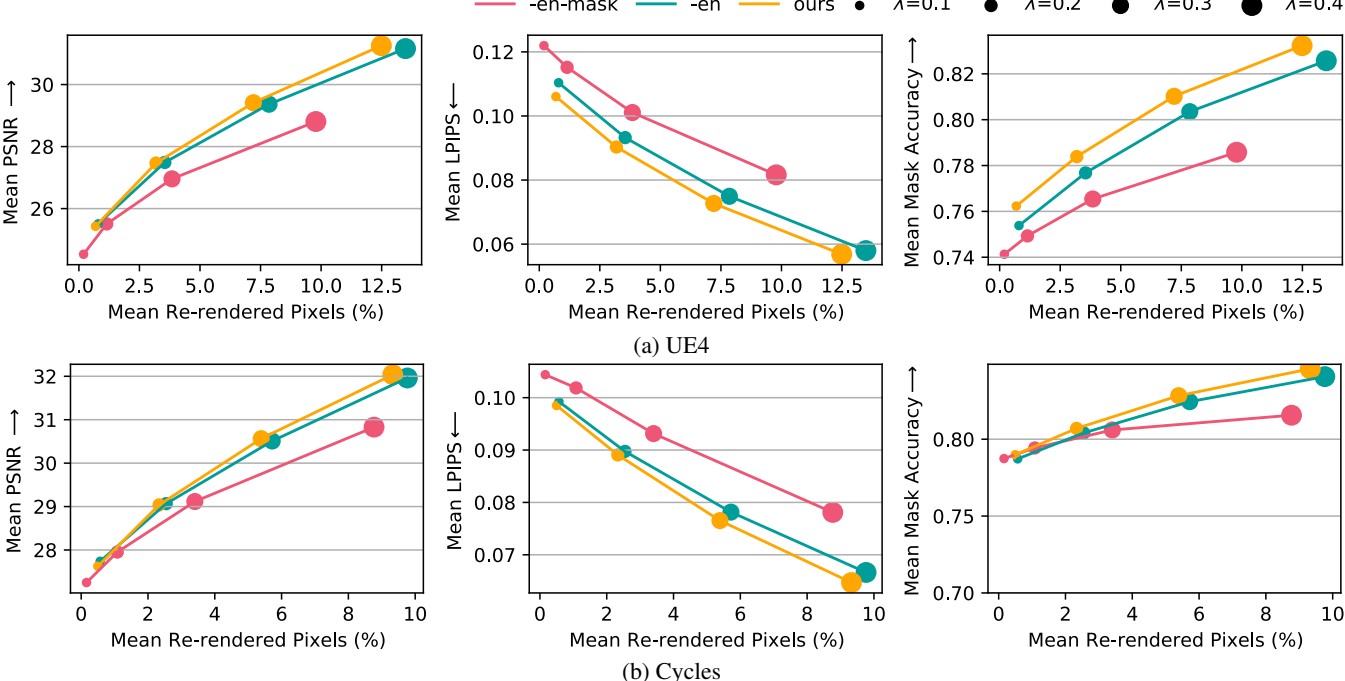

Figure 7: Ablation studies on our UE4 and Cycles testing sets. "-en" denotes our results without intensity enhancement and "-en-mask" denotes our results without intensity enhancement and without confidence mask.

Table 1: Comparison with video frame prediction methods. To ensure a fair comparison, we only train our network to predict one future frame using the $l_1$ loss and generate the prediction results without re-rendering unreliable pixels in this test.

| Method | UE4 | | Cycles | |
|---|---|---|---|---|
| | PSNR | LPIPS | PSNR | LPIPS |
| SDC2D [35] | 27.907 | 0.0561 | 31.065 | 0.0491 |
| SDC2D-GTflow [35] | 28.030 | 0.0573 | 31.033 | 0.0512 |
| MCNet [44] | 23.401 | 0.2363 | 24.943 | 0.2386 |
| VoxelFlow [53] | 25.195 | 0.0898 | 28.733 | 0.0896 |
| ImprovedVRNN [7] | 28.039 | 0.1241 | 31.009 | 0.1055 |
| Ours | 28.618 | 0.0557 | 31.534 | 0.0520 |

optical flows are only used to predict the first future frame in both our method and SDC2D-GTflow. We train all these methods using our training set and validation set. As reported in Table 1, our method achieves better quantitative results than those video frame prediction methods with a large margin (>0.5dB). Compared to those video frame prediction methods, our method also generates qualitatively better results, as shown in Figure 6.

## 5.2 Ablation Study

We examine the effect of two components on our future frame synthesis quality. The first is our intensity enhancement network that compensates for the intensity difference. The second is the optical flow confidence mask, which is used to as a part of the input to the backward flow estimation network. When estimating the first future frame, this map is calculated by assessing the quality of the optical flow generated by the rendering engine. When predicting more future frames, it is predicted using the confidence mask estimation network as described in Section 4. In our ablation studies, we compare three versions of our method, our full method (ours), our method without intensity enhancement (-en), and our method without intensity enhancement and without inputting the confidence

map to the backward flow estimation network (-en-mask). As shown in Figure 7, these two components both help our method predict future frames.

## 5.3 Discussions

We observed that our future frame synthesis method still cannot handle several challenging scenarios. As shown in Figure 8 (a), our method as well as other methods fail to preserve the fine structure (the silver thread). Our method also cannot deal with significant view-dependent effects. Figure 8 (b) shows such an example where the reflection in the mirror is not predicted accurately in the pointed area indicated by the orange arrow. As also shown in Figure 6 where we do not re-render bad pixels, future frame synthesis in general produces errors along object boundaries, mostly due to occlusion.

It takes our PyTorch implementation about 0.02 seconds to predict a $1024 \times 1024$ frame using one Nvidia 3090 GPU. The reported duration includes all the stages of our method except running the rendering engine to replace the unreliable pixels with rendered pixels. The peak GPU memory is about 5400 MB.

In the future, we would like to extend our work by utilizing G-buffer data like many other recent rendering papers [8, 13, 14, 20, 21]. We hope to overcome existing artifacts by adopting a more powerful neural network. We would also like to optimize our network architectures to further speed it up.

### 5.3.1 Use cases

We envision two different usage scenarios i) network deployment on the same system to predict subsequent frames to reduce rendering compute needed and ii) in usages such as cloud gaming, where we may need to predict subsequent frames due to network inconsistencies or frame drops. In case of deploying the network concurrent with the rendering, the confidence mask can be used to selectively re-render the pixels. Based on the dataset and content, the rendering engine will need to re-render a magnitude less pixels compared to re-rendering the whole frame. For high quality rendering such as ray or path traced content, neural frame prediction could be applied

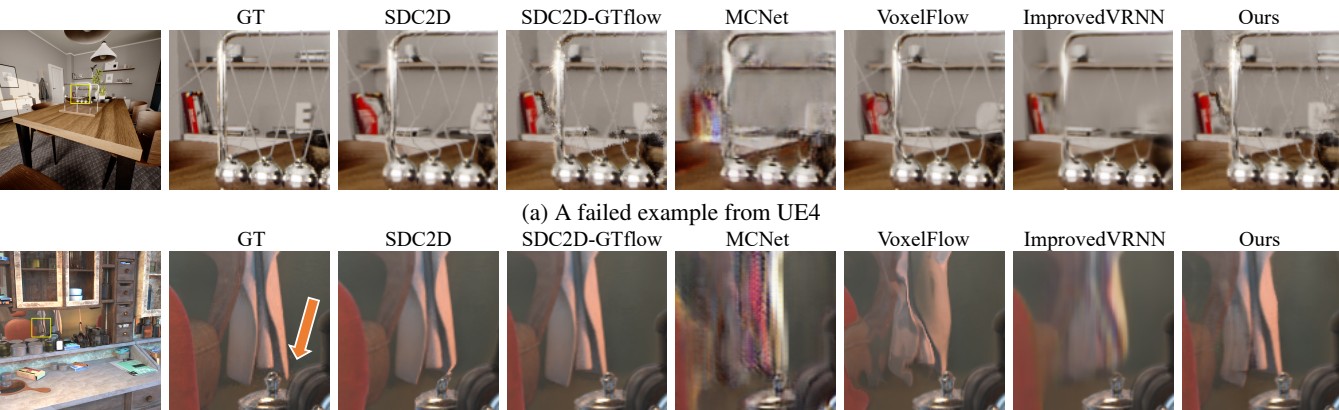

(a) A failed example from UE4

(b) A failed example from Cycles

Figure 8: Failure examples for future frame prediction.

to extrapolate a majority of pixels in the frame, and the limited ray-tracing budget available could be focused on the pixels as determined by the confidence mask. Even for real-time content such as rasterized games, this method could be applied in addition to the geometry processing step to reduce the number of pixels that need to be shaded in the pixels shader stage of the pipeline. Given the limited dependency on the input buffers (2 past RGB frames), compared to concurrent work by Gu *et al.* [14], our system needs less memory footprint with output quality tradeoff.

With gaming and interactive content increasing moving to cloud-based delivery, we envision neural frame extrapolation to be helpful in delivering a compelling user experience across varying compute and network conditions. For example, in cloud gaming scenarios, the game is rendered on a server and streamed to the client over public networks, while the user input is delivered to the server to render the next frame. Given the limited bandwidth and network congestion, dropped or stalled frames could lead to game stutters and unplayable experience. As most client systems do have specific capabilities to run deep neural networks, it is possible to use the neural network engine in the client system to infer the future frame using our approach, while its rendering engine could use the confidence map to re-render limited amount of the frame. One drawback of such a method is that the instance of the game of rendering content will have to be running simultaneously in both the client and the cloud (although the client only renders a small number of difficult-to-predict pixels), with any updates being reflected across both. An alternative approach would be to use the extrapolated frames directly without re-rendering of the lower confidence pixels, with the next rendered frame following shortly thereafter. User studies to gauge the effect of re-rendering v/s utilizing the extrapolated pixels (i.e: not using the confidence map) are future work.

## 6 CONCLUSION

In this paper, we described a method to speed up Monte Carlo rendering algorithms by solving it as a frame prediction problem. To get high quality results, we designed a neural network that not only predict flows to warp future frame, but also predict masks to efficiently rendering pixels that are hard for frame prediction problem. We also propose an enhancement part to strengthen our predictions.

## ACKNOWLEDGMENTS

Models in Figure 1 are from Blender Open Movie Spring [1]. Models in Figure 2 are from *Mixamo*; Denys Rutkovskyi, KK Design, PolyPixel, PROTOFACTOR INC, Pasquale Scionti and Epic Games in *UE Marketplace*; Blender Open Movie Agent 327 [1]; and Nvidia ORCA dataset [27]. Models in Figure 3 are from Renderpeople and Epic Games in *UE Marketplace*. Models in Figure 5 are from *Mixamo*, bought in *UE Marketplace* and from Blender Open Movie Agent 327 [1]. Models in Figure 6 are from Epic Games in *UE Marketplace*, Blender Open Movie Spring [1] and Nvidia ORCA dataset [27]. Models in Figure 8 are from Pasquale Scionti in *UE Marketplace* and Blender Open Movie Agent 327 [1]. This project is supported by a gift from Intel.

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
