# OpenReview forum: "Future Frame Synthesis for Fast Monte Carlo Rendering"
_graphicsinterface.org/Graphics_Interface/2022/Conference — GI 2022_

### Official Review · Reviewer_3aFp · 2022-04-12
**Lack of comparisons and improper motivation**

**Rating:** 4
**Confidence:** 4

**Review:**

The paper presents a temporal reprojection technique that makes the rendering process fast. The high-level idea is to train a neural network that predicts a backward optical flow and compensation term for temporal shading changes. Also, the network guides a rendering engine to re-render some image areas through an estimated confidence map that indicates the per-pixel reliability of the reprojected pixel colors.

This new method is compared to a forward reprojection technique and existing general reprojection techniques for videos, and the demonstrated results indicate that this technique is better than the tested alternatives visually and numerically.

The primary concern is that it is unclear why it is challenging to use G-buffers. The temporal reprojection has been widely exploited for real-time rendering scenarios, and a common way is to exploit G-buffers (e.g., depth information) when computing the optical flow. Extracting the depth information can be quite efficient for rendered sequences, and one can easily estimate (or compute) the per-pixel optimal flow, e.g., using the depth and camera information (e.g., 4 x 4 viewing matrix). Then, this flow can be directly employed for a backward temporal reprojection. Also, one can compare the depth value at a pixel in the current frame with the value at the corresponding pixel in the previous frame to measure the reliability of this reprojection. If the difference in the two depth values is larger than a chosen threshold, this pixel can be in a dis-occluded area. As a result, one can re-render only this pixel.

If the application of this method is a general video, it does make sense to assume that extracting G-buffers can be challenging. However, this paper addresses rendered image sequences, and thus, the standard technique above should be directly compared to motivate the current design choice, i.e., predicting optimal flows and conducting a backward reprojection without G-buffers.

It is hard to evaluate this proposed idea and neural framework without this comparison (and proper motivation), and thus I cannot recommend the acceptance of the paper.

---

### Official Review · Reviewer_ntWz · 2022-04-13
**Good paper**

**Rating:** 6
**Confidence:** 5

**Review:**

The paper presents a method for fast Monte Carlo temporal rendering. The basic idea is to construct a future frame by using the optical flow from the future frame to the current frame to determine reusable pixels. Monte Carlo ray tracing can be done on the remaining pixels to complete the future frame. To improve robustness, the authors proposed to predict the optical flow using a neural network, together with extra maps including the confidence map and mask. The paper comes with a new animation dataset that can be used to train the proposed network. The experiments show that the proposed method is effective and slightly outperforms the baseline methods.



**Clarity**

The paper is relatively well written. Overall I enjoyed reading it.

Some minor issues of the presentation:

- It is hard to see the difference between the images in the teaser. Its caption can be refined to demonstrate the motivation of the brightness change map better.



**References**

Adequate.

As the method has quite some steps, reproducing this paper might need tremendous effort. I encourage the authors to make their code and data publicly available.



**Technical**

The proposed method largely depends on the optical flow, which is a fundamental problem in computer vision. Training optical flow networks using a neural network is well explored. There exist public synthetic datasets in computer vision such as Sintel to train such networks. I wonder why a new dataset is required in this case.

The new dataset is constructed by imposing an animated character on a background. This is a common scenario in animation, and so in general the dataset is reasonable. I think a potential reason for creating this dataset is for Monte Carlo rendering scenario, which is fine. In that case, I expect to see some comparisons of the proposed optical flow network on Sintel dataset to understand the performance on this public benchmark.



**Experiments**

The improvement is marginal, demonstrated by both PSNR and LPIPS metric in Table 1. I think more explanations about the current performance is preferred. For example, there are a few animation sources in the tested scene, e.g., from the background due to camera movement, and from the animated character. It would be good if the authors can analyze to see which is a source of large errors, which causes the marginal improvement. From the visualization in Figure 6, it seems the main error is due to occlusions.



**Justification**

This paper presents a practical solution to the important problem of temporal rendering, which would be applicable in the computer graphics industry. The authors demonstrate that some improvement can be achieved, while the margin is not significantly large. Given its usefulness and technical soundness, I am happy to accept this paper to GI.

---

### Official Review · Reviewer_9Kop · 2022-04-14
**Good rendering research paper with interesting ideas.**

**Rating:** 7
**Confidence:** 4

**Review:**

This paper proposes a novel re-rendering technique based on predicting backward flow with a neural network for rendering. The novelty of this work, aside from not requiring G-Buffer, is the two additional networks trained in an end-to-end fashion. One network will predict a (binary) mask to identify reprojected pixels that produces the highest error. All pixels identified by the mask will be recomputed, leading to a significant improvement in terms of image quality. The other network predicts the brightness change between two consecutive frames. This brightness correction objective is to correct the change in intensity of the warped pixels (from the previous frame) due to changes in geometry. An ablation study is performed to check how important are these two additional networks. Moreover, empirical evaluation is conducted against (forward) reprojection baselines (reproj-nn and reproj-blend) and other ML-based reprojection algorithms. Compared to these prior techniques, the proposed technique shows better image quality.

The paper is well written and easy to understand. I like the idea of predicting the backward optical flow with confidence maps and brightness changes. I think "Brightness" is not the most accurate word here, as it is visible in Figure 3 that B-maps are RGB. It will be better to call these maps "intensity difference" or "delta intensity." I wish that the difference between forward vs. backward warping has been further discussed. Indeed, forward wrapping can generate holes inside the warped image, resulting in major artifacts. However, these pixels can be directly added to the mask to be recomputed by the rendering engine. I would like to see more explanation on this. Moreover,  a better justification for why only one image sequence is used for validation must be provided. Finally, I suggest removing the weights in equation (2) as there are all equal.

The reference list seems rather long, and some not relevant work in denoising (for rendering) is listed. It will be better to focus on (recent) techniques that reduce rendering time using reprojected information. Here are some relevant references in rendering:
- "Neural Temporal Adaptive Sampling and Denoising", Hasselgren et al., Eurographics 2020
- "Denoising with kernel prediction and asymmetric loss functions", Vogels et al., SIGGRAPH 2018
- "Fast temporal reprojection without motion vector", Hanika et al., JCGT 2021

Please use this reference list to find other related work in this area.

In addition, it will be good to discuss the proposed technique compared to adaptive sampling techniques (e.g., "Deep Adaptive Sampling for Low Sample Count Rendering"). Indeed, in adaptive sampling, we often predict a (non-binary) map to know which pixels need to be rendered more. Adaptive sampling approaches are quite related to using a mask to only re-render a part of the image. On a side note, please avoid using a long list of references without quickly explaining the paper (or the group of papers).

The proposed approach is technically sound. However, I do not fully agree with the authors that it is good not to have G-buffer information. Indeed, generating such information is quite cheap (compared to computing shading and lighting) and brings a lot of information about scene geometry (e.g., depth), material (e.g., normal, diffuse color), and lightings (e.g., with albedo demodulation). It is why these auxiliary buffers are almost always used in prior works in rendering. I understand the use case described in section 5.3.1. But assuming that the client might re-render some pixels is (might be?) much more costly compared to generating this G-buffer information. However, this G-buffer information might not be accurate in some situations due to participating media presence or large motion blur/depth of field. In this case, the proposed technique should be quite competitive compared to techniques that heavily rely on G-buffer information.

The comparison results look excellent. However, a major catch here is that the proposed technique re-renders some pixels during the animation (gain more information) compared to all other techniques. This gives the proposed technique an unfair advantage, especially as removing this ability has a huge impact on the proposed technique's performance (Figure 7 -- Ablation study). This is quite problematic as no actual figures about the cost of re-rendering these pixels are never mentioned inside the text (maybe quite challenging to compute?). It will be essential to bring this information to the reader if the manuscript is revised.

In conclusion, I like the idea of predicting the backward flow and trying to improve the results with deep learning. Even if the comparison is quite unfair to other techniques compared inside the paper, I think the paper has interesting ideas to propose. This is why I am for accepting this paper in GI 2022.

---

### Decision · Program_Chairs · 2022-04-17

Accept